# A dataset for the study of identity at scale: Annual Prevalence of American Twitter Users with specified Token in their Profile Bio 2015–2020

**Jason Jeffrey Jones** *

Department of Sociology and Institute for Advanced Computational Science, Stony Brook University, Stony Brook, NY, United States of America

* jason.j.jones@stonybrook.edu

## Abstract

Personally expressed identity is who or what an individual themselves says they are, and it should be studied at scale. At scale means with data on millions of individuals, which is newly available and comes timestamped and geocoded. This work introduces a dataset for the study of identity at scale and describes the method for collecting and aggregating such data. Further, tools and theory for working with the data are presented. A demonstration analysis provides evidence that personal, individual development and changing cultural norms can be observed with these data and methods.

## Introduction

Personally expressed identity should be studied at scale. Personally expressed identity is who or what an individual *themselves* says they *are*. "At scale" means with data on millions of individuals. Such data happens to be attainable, and it comes timestamped and geocoded–meaning temporal trends and geographic comparisons are readily analyzable.

This article introduces a publicly available aggregated dataset for the study of temporal trends in personally expressed identity. Named "Annual Prevalence of American Twitter Users with specified Token in their Profile Bio 2015–2020," the data provides a measure of the popularity of words chosen by Twitter users in the United States of America for inclusion in their profile biography. It represents a first step toward a goal of a multinational, multilingual, continuously updated data pipeline for the study of identity at scale.

In addition to the dataset, this article contains four further contributions:

1. Discussion of a general method for collecting personally expressed identity at scale.

2. A theoretical framework for understanding data of this form and type.

3. An analysis using the dataset to illustrate how temporal trends may be observed.

4. Brief instructions for an online tool that makes the data accessible without computer programming.

**Data Availability Statement:** All database and analysis code files are available from the Open Science Foundation (https://osf.io/guah5/ and https://osf.io/rwxec/).

**Funding:** This material is based upon work supported by the National Science Foundation under grant IIS-1927227. The author thanks Stony Brook Research Computing and Cyberinfrastructure, and the Institute for Advanced Computational Science at Stony Brook University for access to the high-performance SeaWulf computing system, which was made possible by a $1.4M National Science Foundation grant (#1531492). The funders had no role in study design, data collection and analysis, decision to publish, or preparation of the manuscript.

**Competing interests:** The author has declared that no competing interests exist.

## Prior work

This work follows from the practices of culturomics and the study of self using so-called "Who-am-I" instruments. Culturomics [1] is the quantitative study of cultural trends with large datasets. Michel et al. introduced the term and studied the language of books printed in English from 1800–2000. They demonstrated linguistic change over time (e.g. "burned" over-taking "burnt" in usage) and common patterns in collective memory (e.g. celebrity names generally rose rapidly from obscurity to peak usage then experienced a slow decline). The current work is based on the founding idea of culturomics: language use reveals much about the minds of authors and the collective consciousness of the society in which they are embedded. Large digitized text corpora provide the luxury of studying these objects with both breadth and precision.

Studying identity with language is not new. It was not new five decades ago, as Spitzer et al. [2] lamented: "A perusal of Wylie's The Self Concept [3] discloses the existence of no fewer than 100 instruments, only a small minority of which have seen repeated use. It seems that every student of the self-concept, either because of dissatisfaction with existing instruments or the choice of research problems, has contributed at least one additional device." Many of these instruments were based on free-response prompts for self-descriptive language, sometimes collectively called "Who-am-I" instruments. The most frequently used of these came to be the Twenty Statements Test or TST [4]. The TST is comprised of twenty prompts of the same form: "I am ______." This simple format is easy to administer and elicits rich language data. However, it suffers from the limitations of any administered instrument. It is difficult and expensive to collect large response sets, especially longitudinally.

The trend of tweaking the "Who-am-I" format continues; in 2021 another extension of the TST was proposed [5]. This extended instrument demonstrated there are aspects of identity that do not come immediately to mind and therefore are underrepresented in TST results. Probing along specific dimensions (such as aspirations) predictably brought out more responses along the probed dimensions.

## Advantages of the current research

"Who-am-I" instruments are all limited in the scale they may be deployed and by the artifacts of responding to researcher-determined items in an unnatural environment. The current work has the advantages of studying personally expressed identity text relatively unobtrusively, at scale and longitudinally.

## Materials and methods

### Longitudinal Online Profile Sampling (LOPS)

LOPS stands for longitudinal online profile sampling. LOPS is a method for studying the choices people make when describing themselves with words. Let's call the output of this act: personally expressed identity. It is personal–the individual is describing themselves. It is expressed–these are words the individual emits, where others might see them. And it describes identity–the explicit purpose of the text is description of the author.

**The profile bio.**   Personally expressed identity may be observed in the language of profile bios. The profile bio is a short text written by an individual to describe themselves. Bios are a feature of profiles on many social networking sites, including Facebook, LinkedIn and Reddit. This work is focused on Twitter. On Twitter, the bio appears on the profile page just below the picture and name of the user. When the user first creates an account, they respond to the

prompt: "Describe yourself in 160 characters or less" to create the bio. The user may update their bio at any time.

To perform LOPS, one samples bios from online profiles as often as possible. For example, one might take a snapshot of each tweet from one user, including the user's profile at the time. This allows a series of time-stamped self-descriptions to be assembled to form a longitudinal picture of that user's personally expressed identity. From such series, compiled for many users (multiple millions of US Twitter users in the present work) one can draw inferences about temporal trends in aggregate identity.

LOPS is a general method that could be instantiated in a myriad of ways. The present work is focused on Twitter users in the United States, however, there are many other potential sources of personally expressed identity data. LOPS requires only repeated sampling of short-text self-descriptions. Social media profiles are the obvious first target but not the only. Contributor bios can be found online elsewhere–for instance, on blogs and Wikipedia user pages. Job-seekers often include a brief description of themselves on their resume. Journalists and other authors write brief bios that accompany their work. Moving beyond written personal profiles, people engage in self-description in many forms. Probably everyone has performed a formal introduction of themselves in a meeting. At a party, self-introductions are a social norm. All of these examples are modes of personally expressed identity, and contain rich language data that can be studied with the same techniques as applied in this manuscript.

## The dataset: Annual Prevalence of American Twitter Users with specified Token in their Profile Bio 2015–2020

Briefly, the dataset comprises a single table containing the incidence and prevalence per year of American users who chose to use a linguistic token in their Twitter biography. The incidence and prevalence were computed separately for two samples, one cross-sectional and the other longitudinal. The cross-sectional sample included all users observed tweeting at least once at any time 2015–2020. The longitudinal sample includes only those users who were observed at least once per year in each and every year (i.e. 2015, 2016, 2017, 2018, 2019 and 2020).

## Constructing the dataset

The dataset was derived from a random sample of tweets. From 2015 through 2020, I used the Twitter Streaming API [6] to observe a random 1% sample of all tweets. I filtered the tweets to likely US users by examining the profile location field. The location text is entered by the user, and I filtered out texts that were likely not in the US (e.g. "London, UK") and selected in texts that indicated a US location (e.g. state names and formal abbreviations). Fig 1 contains a diagram of the dataset construction process.

**Cross-sectional sample.** The cross-sectional sample was constructed first. As described above, a random sample of tweets provided a stream of profiles (including bios) that were filtered to likely US locations. For each year, only one bio per user was sampled. The bio was chosen at random from all observations of a particular user's tweets. Thus, whether a user was observed tweeting once or a hundred times, exactly one bio is recorded per user per year. About 10 million unique users were observed per year. Exact counts can be found in Table 1.

The cross-sectional sample is valuable, because it allows for inference concerning population-level change. We may observe how the set of active users within each year described themselves. If a token rose in prevalence in this sample, it may have been due to a new influx or exodus of users. It could have been due to usually dormant users becoming momentarily more active. There could be many causes for the relative prevalence of tokens to change across

**Fig 1. Process for creating datasets.** This flowchart describes the process of creating the Cross Sectional and Longitudinal datasets.

cross-sections. This sample allows for temporal trends in personally expressed identity to be analyzed in a large set of contemporaneously active users.

**Longitudinal sample.**   The bios in the longitudinal sample are those for users observed in every year 2015 through 2020. There are over a million accounts that meet this criterion.

**Table 1. Counts of unique US users per sample and year.**

| Sample | Unique User Count |
|---|---:|
| Cross-sectional 2015 | 8,564,955 |
| Cross-sectional 2016 | 10,227,688 |
| Cross-sectional 2017 | 10,638,679 |
| Cross-sectional 2018 | 10,310,854 |
| Cross-sectional 2019 | 9,817,008 |
| Cross-sectional 2020 | 10,181,678 |
| Longitudinal 2015–2020 | 1,353,325 |

Cross-sectional samples contain one and only one profile snapshot per user (chosen at random) from within the named year. The longitudinal sample contains only those users observed in every year.

Another way to imagine the longitudinal sample is as the intersection of users from the six annual cross-sectional samples.

The longitudinal sample is valuable, because it allows for inference concerning individual change. We may observe how the same set of users described themselves at six points in time over half a decade. If a token rose in prevalence in this sample, we may safely conclude that more individuals added the token than removed it. Personally expressed identity may be observed changing within individuals. In this sample, about 52% of users edit their profile in any given year. 88% of users edited their bio at least once from 2015 through 2020.

To illustrate the sampling processes, assume we observed User A tweet once in 2016 and three times in 2018. User A's profile biography from 2016 was included in the cross-sectional sample for 2016. One of the three tweets from 2018 was chosen at random, and User A's profile biography accompanying that tweet was included in the cross-sectional sample for 2018. User A was not included in the longitudinal sample, because there were some years in which we observed no tweets for User A.

**Annual counts by token.** The dataset does not consist of bios. Instead, it provides the annual incidence and prevalence of users with each token within their bio for each sample. Incidence and prevalence were computed by tokenizing each bio and tallying per year the number of unique users choosing to include a token. Formally, prevalence is defined as:

$$Prevalence = 10,000 * {Count\ of\ Users\ with\ Token}/{Total\ User\ Count}$$

The ratio of users with the token to total users is multiplied by 10,000 because, of course, most bios do not contain most tokens, and it is much easier for humans to think in whole numbers than small fractions or decimals [7]. It is important to be clear that reported counts are always counts of users and never words. A bio that reads "token token token token" counts as one user whose bio contains "token," even though the bio text happens to contain "token" four times.

Table 2 contains several rows from the data as an illustrative example. The table contains every row for the token "maga." From the Prevalence column, one can observe that accounts whose bio contained "maga" grew in prevalence every year in both the cross-sectional and longitudinal samples. A token must have prevalence greater than or equal to 1 for a row to be included, so one can infer that the prevalence of "maga" was less than 1 for both samples in 2015. Below I describe each column in the data.

*Token*. Token contains a character string obtained by tokenizing a Twitter bio. A "Twitter bio" is the name I use for the "description" field of the "User" object in the Twitter API. Twitter bios contain users' responses to the prompt: "Describe yourself in 160 characters or less." The

**Table 2. Example rows of annual token data.**

| Token | Year | Prevalence | Numerator | Denominator | Sample Type |
|---|---|---|---|---|---|
| maga | 2016 | 6 | 5,656 | 10,227,688 | cross |
| maga | 2017 | 24 | 25,791 | 10,638,679 | cross |
| maga | 2018 | 53 | 54,409 | 10,310,854 | cross |
| maga | 2019 | 63 | 61,980 | 9,817,008 | cross |
| maga | 2020 | 70 | 71,617 | 10,181,678 | cross |
| maga | 2016 | 4 | 544 | 1,353,325 | longi |
| maga | 2017 | 17 | 2,249 | 1,353,325 | longi |
| maga | 2018 | 32 | 4,282 | 1,353,325 | longi |
| maga | 2019 | 39 | 5,211 | 1,353,325 | longi |
| maga | 2020 | 40 | 5,478 | 1,353,325 | longi |

This table contains all the data for the token "maga" within the dataset *Annual Prevalence of American Twitter Users with specified Token in their Profile Bio 2015–2020*. Prevalence was calculated as Numerator / Denominator * 10,000. Numerator was the incidence of accounts within the named sample, and Denominator is the total accounts in that sample. The cross-sectional sample is marked with "cross" in the Sample Type column, and "longi" refers to the longitudinal sample. One can infer from a lack of a data row that Prevalence was less than 1, because only those rows with Prevalence greater than or equal to 1 were included.

tokenization is a simple string split on the regular expression \b|\s+, which matches word boundaries or whitespace. Many tokens are words like one would find in the dictionary, but not all. Abbreviations, numbers, and other non-words also appear.

*Year*. Year contains the four-digit year, one value from the range 2015 through 2020.

*Prevalence*. Prevalence contains a useful measure of the popularity of a token among US users of Twitter. Precisely, the value of Prevalence in the dataset is 10,000 * (Numerator / Denominator) rounded to the nearest integer. This should be interpreted as the number of users per 10,000 unique users who had a profile containing the token. Prevalence is convenient, because it avoids small fractions and decimals. Most tokens are NOT used by most users, so all incidences must be small compared to the total user count. Tokens with a prevalence less than 1 are not included in the dataset. This removes a large set of low-frequency tokens that are likely uninteresting (e.g. obscure references, idiosyncratic misspellings and unique identifiers).

*Numerator*. Numerator contains the count of users where Token was found within their bio. This is the incidence of the token within the sample, but for most applications prevalence is likely more useful than incidence.

*Denominator*. Denominator contains the total count of unique users within a sample. The denominator will always be exactly the same within a Year and a Sample Type. For the longitudinal sample, the number is exactly the same for every year, because it is the number of unique users who are present across all years—i.e. the intersection of all the 2015–2016 cross-sectional samples.

*Sample Type*. Sample Type contains one of two possible values: cross or longi. Cross means the data come from the cross-sectional sample. The value longi means the data come from the longitudinal sample.

The dataset can be downloaded in its entirety at https://osf.io/guah5/. There one will also find a README file which describes each column and the procedure by which the data was generated. A demonstration R script loads the data and visualizes a few series to further familiarize the user with the data's structure.

## Theoretical framework: Words as stocks, prevalence as price

A stock market is the organizing metaphor in my mind when I ponder this data. In the metaphor, linguistic tokens (i.e. words and word-like things such as abbreviations, hashtags and

emoji) correspond to stocks. In other words, one could think of the token "maga" as MAGA–a ticker symbol for this particular element of personal identity. It represents that element in the same way AAPL represents Apple, Inc on the NASDAQ stock exchange.

The prevalence of a token is a direct measure of its popularity of use. Indeed, it is determined by the count of individuals who have chosen to describe themselves with that token. Thus, prevalence corresponds (metaphorically) to price; the more demand for a stock, the higher its price, and the more demand for a token, the higher its prevalence.

This is a simple metaphor. It is a one-to-one mapping of two concepts onto two others. But it is useful for several reasons. First, it transforms what is likely unfamiliar–"prevalence of linguistic tokens"–to something everyday. Everyone has heard of a stock price going up or down; the directions of the major indices are reported every evening in the news. Second, the framework suggests multiple analytic approaches. In stock trading, "technical" traders look at charts, consult historical time series and make predictions with only modest regard for what a company is inherently "worth." One can do the same with token prevalence. Charts based on the present data will give a sense for "momentum" or "meme" [8] tokens that are changing quickly in prevalence.

"Fundamentals" traders, on the other hand, are not so much concerned with current price or price history, but instead on what a stock's price should be, or what it shall be once momentary, transient volatility settles out. A fundamentals analysis would predict that religious identifiers should decrease in prevalence among Americans as long as the long-term trend of decreasing religiosity [9] continues. A fundamentals perspective would find it confounding that Americans are rapidly adding political party identifiers to their bios [10] while survey results show Americans' political party affiliations are changing slowly and toward increasing identification with the Independent label rather than major party affiliation [11].

In the same way that stock markets provide information about what market participants find valuable monetarily, token prevalence estimates reveal what social media users find valuable in self-presentation. Responding to economic conditions, market participants "rotate" investments–for example, selling technology company stocks and buying utility provider stocks. Similarly, individuals respond to social conditions and may rotate out of pop culture references into political activism, for instance. Just like stock prices, token prevalences update every day based on the actions of many individuals. Tracking these numbers provides both a sense of current perceptions of value and the historical path to that value. Brave or foolhardy souls can then do their best to predict future values.

## Results and discussion

### Analysis: Temporal trends in token prevalence–winners and losers

In stock markets, there are winners and losers. Winners increase in price and losers decrease–sometimes slowly and steadily and sometimes vertiginously. As a demonstration application, here I use the dataset to examine temporal trends in token prevalence, and highlight those tokens whose popularity has dramatically changed.

**Descriptive analysis.** First, we should recall the primary measure of interest–prevalence. Prevalence (defined previously in Equation 1) is the ratio of users who describe themselves using a token to total users multiplied by 10,000. Fig 2 contains a histogram for each year of data from the longitudinal sample depicting the distribution of token prevalences. Distributions for the cross-sectional sample are very similar and appear in the S1 File. In the entire dataset, 17,765 unique tokens meet the criterion of prevalence greater than or equal to one; within any one year the number of unique tokens is between twelve and fifteen thousand. From the histograms, one can see clearly that there are a small number of tokens of high user

## Prevalence Distribution, Longitudinal Sample, by Year

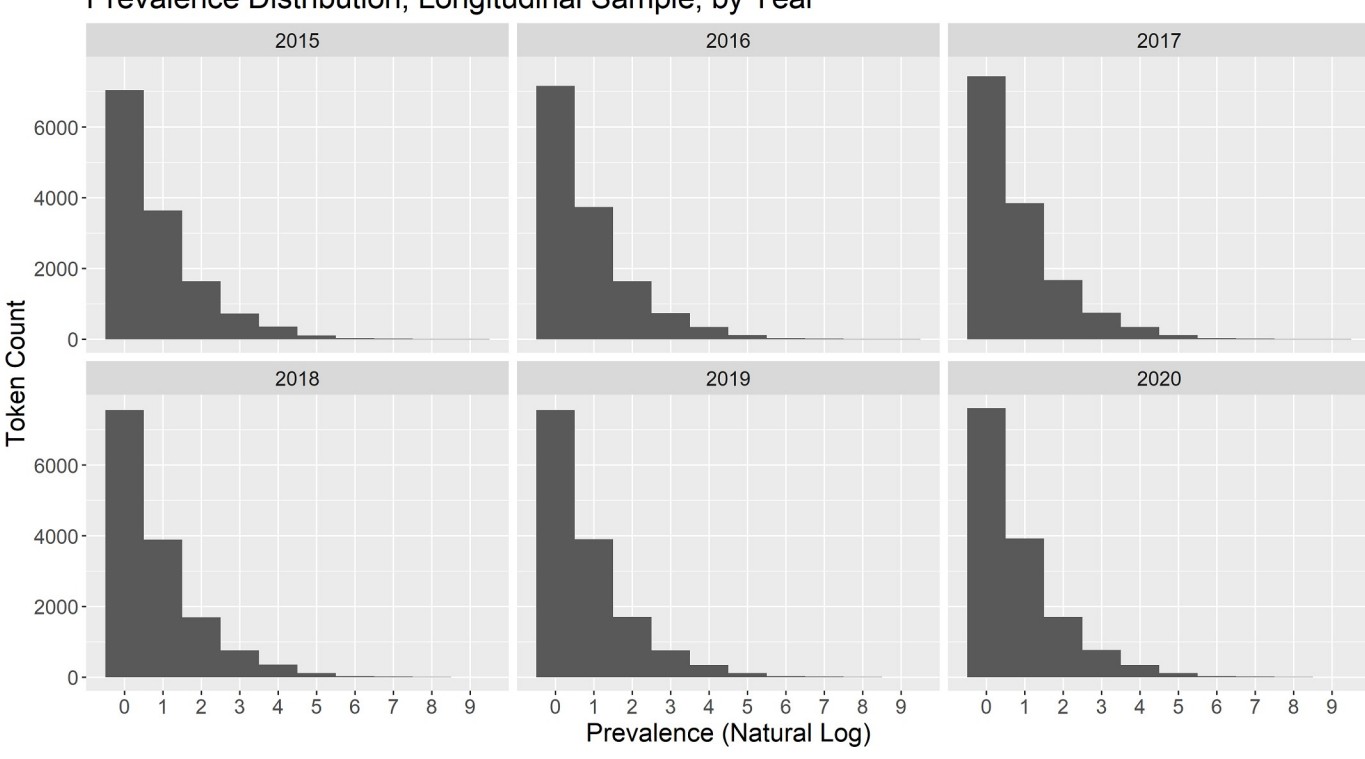

**Fig 2. Token prevalence distributions per year for the longitudinal sample.** Note that the x-axis is on a log scale. There are a small number of high-prevalence tokens and large numbers of low-prevalence tokens. This accords with general expectations of word usage.

prevalence and many tokens of low user prevalence. The poweRlaw package in R [12] implements the suggested procedures of [13] for characterizing heavy-tailed distributions such as this one. These techniques yielded the conclusion the log-normal distribution is a better fit than the alternatives of a power-law, exponential or Poisson. The best-fitting log-normal has parameters $\hat{\mu} = -6.21$ and $\hat{\sigma} = 3.31$. Best fits of each distribution to the data are plotted in the S1 File.

Table 3 contains the 20 "most-surprisingly common" tokens from the longitudinal sample in 2020. Most-surprisingly common is defined as below. Based on prevalence, each token is assigned a Prevalence Rank (PR). The most prevalent token receives a PR of 1, the second-most 2 and so on. For the same tokens, word frequency estimates [14] from the Corpus of Contemporary American English [15] were used to estimate Word Frequency Rank (WFR) for English language usage generally. The level of "surprise" for seeing a token in a bio is the ratio of WFR to PR. When the value is close to 1, the token is used about as often in language generally as it is in profile bios. When the value is less than 1, the token appears less often in bios than one would expect. When the value is greater than 1, this indicates a low-frequency word has high prevalence in profile bios. The Spearman correlation of WFR and PR was $r = 0.54, p < 0.001$.

Some words appear at expected rates. The tokens "and," "at" and "next" have a WFR to PR ratio of exactly 1. Other tokens appear in bios at surprisingly high rates. The token "fan" has a ratio over 50. Users defined themselves based on things they liked. "Lover" and "enthusiast" were rare words that many users incorporated into their bios, with ratios of at least 91 and 61 respectively. (The two words do not appear in the top 5050 words in COCA, so their WFR can

**Table 3. The 20 most-surprisingly common tokens from the longitudinal sample in 2020.**

| Token | Prevalence | Prevalence Rank (PR) | Word Frequency Rank (WFR) | WFR / PR |
|---|---|---|---|---|
| fan | 335 | 40 | 2016 | 50.40 |
| photographer | 97 | 117 | 4907 | 41.94 |
| founder | 125 | 99 | 4059 | 41.00 |
| producer | 128 | 96 | 3758 | 39.15 |
| opinions | 140 | 88 | 3227 | 36.67 |
| advocate | 74 | 134 | 4830 | 36.04 |
| twitter | 171 | 71 | 2528 | 35.61 |
| writer | 271 | 48 | 1692 | 35.25 |
| designer | 84 | 126 | 4206 | 33.38 |
| engineer | 68 | 140 | 4446 | 31.76 |
| consultant | 52 | 156 | 4771 | 30.58 |
| photography | 37 | 171 | 5008 | 29.29 |
| sports | 273 | 46 | 1299 | 28.24 |
| journalist | 45 | 163 | 4525 | 27.76 |
| resist | 29 | 179 | 4881 | 27.27 |
| artist | 181 | 65 | 1769 | 27.22 |
| cats | 34 | 174 | 4673 | 26.86 |
| don | 152 | 81 | 2172 | 26.81 |
| hers | 20 | 188 | 5033 | 26.77 |
| anchor | 22 | 186 | 4915 | 26.42 |

This is a curated list of the 20 tokens that are most surprising in their prevalence when compared to their rate of use in language. Token, Prevalence and Prevalence Rank were calculated from the dataset. Word Frequency Rank was obtained from the 5050 most frequently used wordforms in the 2020 Corpus of Contemporary American English. Only tokens available in both sets are listed in this table.

be assumed to be 5051 or higher. Table 3 does not include tokens without a direct match–and thus a specific WFR–in COCA.)

Creative and professional occupations (e.g. "photographer", "producer," "engineer," "consultant") had high user prevalence compared to the frequency those titles were used in the COCA corpus. Some surprising tokens were likely platform-specific (e.g. "opinions" and "twitter"). On Twitter, many users specify that their account is personal. For instance, in the bio they state "Opinions my own" or "Opinions do not reflect those of <employer>." The most surprisingly missing or disused token in bios was "was"; WFR = 14, PR = 151. Users avoided the past tense in their personally expressed identity text.

**Prevalence winners and losers.** Once we think of tokens as stock tickers, and we have multiple years of prevalence (price) estimates, then a natural fascination emerges for characterizing "winners" and "losers." Winners are tokens whose prevalence has increased over time, and losers are those on a downslope. Because the data contains estimates for every token for every year, we can estimate annual changes in popularity from the coefficient of linear regression.

I estimate the slope for each token using Ordinary Least Squares linear regression. This number has a natural interpretation; it is the expected annual increase in prevalence. A value of 2, for example, implies that prevalence increased by about 2 users per 10,000 each year. It may have increased from 10 to 12 or from 200 to 202. A different analysis might concern increase relative to initial prevalence and result in the two cases above appearing as very different outcomes, rather than the same. Here I focus on absolute change over time.

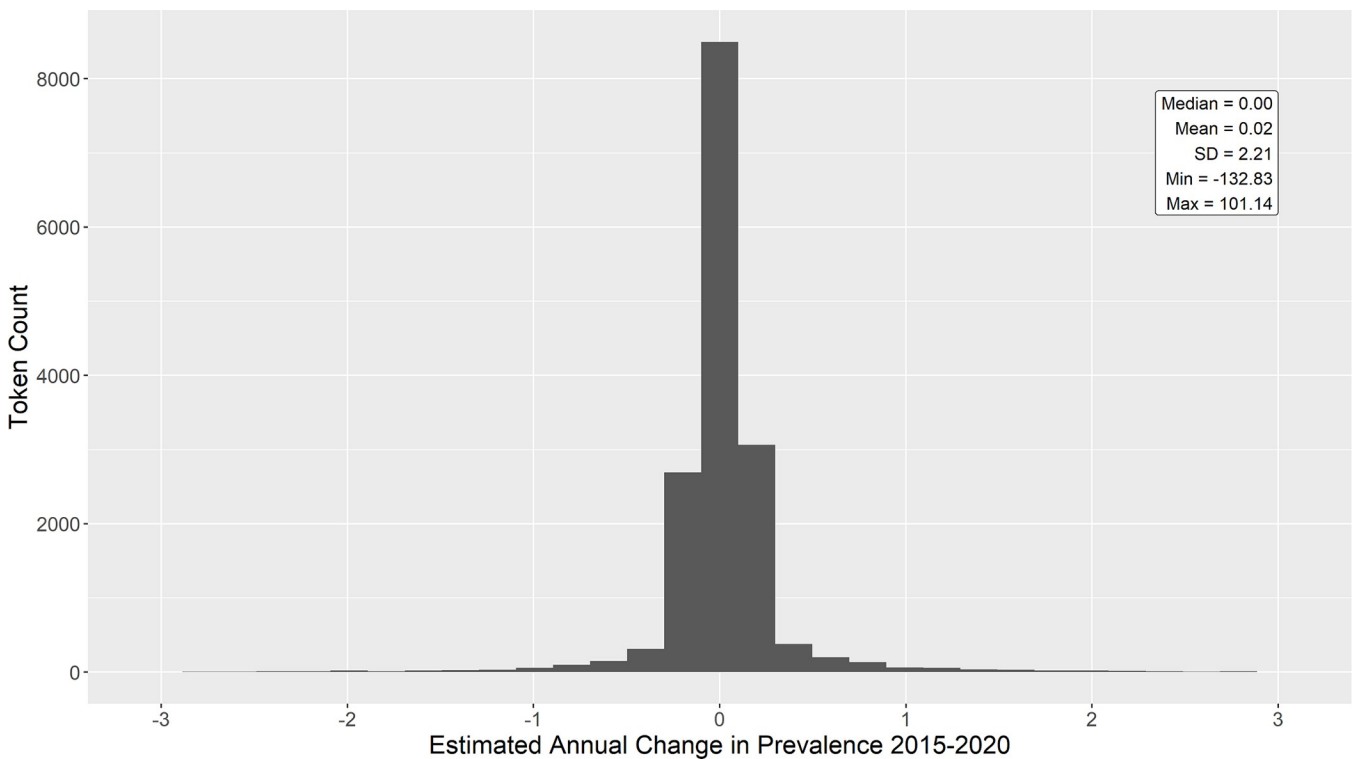

**Fig 3. Distribution of estimated annual change in prevalence for all unique tokens within the longitudinal sample.** Prevalence is stable (i.e. zero change) for many tokens. Note from the min and max annotations that extreme values are present in the data but not pictured here. See Tables 4 and 5 for illustration.

Fig 3 contains a histogram of slopes for all tokens in the longitudinal data from 2015–2020. Prevalence is stable (i.e. zero change) for many tokens. However, there are some tokens that experienced large shifts. Table 4 contains the 20 "biggest winners"–tokens with the largest annual prevalence increases. Table 5 contains the 20 "biggest losers"–tokens with the largest annual prevalence decreases.

In the longitudinal sample winners and losers, one sees evidence of both cultural shifts and personal, individual development. It may at first seem surprising that pronouns occupy the top four winner slots. But this is the result of a developing and spreading social convention: adding pronoun-slash-lists to one's self-description. A pronoun-slash-list is a series of pronouns provided by an individual that indicate the preferred words others should use when referring to the individual in the third person (e.g. he/him). Other winners show the rising prominence of new platforms ("ig," "twitch" and "podcast") and politics ("American flag emoji" and "maga").

The winners and losers lists contain evidence of personal, individual development. Users stopped describing themselves as "student" and became "alum." Years receding into the past dropped in prevalence and were replaced by present and future values. (Some two-digit numbers might also or instead refer to ages, but either way the process is similar.) As they age, people took on new roles–"host" for example–but more momentously "mom" and "dad." In their personal identities, they left vague abstractions behind ("life" and "music") and accumulated experience ("former"). Perhaps they became a bit jaded and lost their innocence; losers included "love," "like," and "red heart emoji."

For comparison, I repeated the analysis for the cross-sectional sample. Fig 4 displays the histogram of estimated slopes for all tokens 2015–2020 for the cross-sectional sample. As

**Table 4. Top 20 winner tokens in the longitudinal sample.**

| Token | Annual Prevalence Change Estimate | Standard Error of Estimate |
|---|---|---|
| she | 40 | 10.63 |
| her | 37 | 10.11 |
| he | 27 | 7.18 |
| him | 26 | 6.77 |
| alum | 18 | 0.27 |
| own | 15 | 1.29 |
| mom | 13 | 0.44 |
| dad | 11 | 0.23 |
| 22 | 11 | 2.63 |
| opinions | 11 | 0.68 |
| American flag emoji* | 11 | 0.91 |
| 21 | 10 | 4.72 |
| ig | 10 | 2.05 |
| former | 9 | 0.25 |
| maga | 9 | 1.09 |
| twitch | 9 | 0.33 |
| podcast | 9 | 0.10 |
| 2020 | 8 | 1.92 |
| 23 | 8 | 0.97 |
| host | 8 | 0.37 |

This is a curated list of the 20 tokens with the largest estimated annual increase in prevalence. Punctuation, individual digits, a small set of stop words (e.g. of) and URL components (e.g. https) have been previously filtered. Emoji results are marked with *.

above, prevalence was stable for most tokens. The most extreme winners and losers are listed in Tables 6 and 7.

Consider that the longitudinal sample (N = 1,353,325 unique users) is a strict subset of the cross-sectional sample (about 10 million unique users per year). The occasional or new users in the cross-sectional sample outnumbered those in the longitudinal sample, but both were present. Therefore, the cross-sectional results represent both change at the individual level and changes in the composition of the in-the-moment active user population. Results must include both cohort and platform-use effects.

The cross-sectional sample results revealed both similar trends and novel token appearances. In the realm of politics, "maga" and "American flag emoji" were winners again and to a larger degree than they were in the longitudinal sample. "Trump" appeared in the top 20 winners, but did not make the cut previously. "BLM" was also on the rise.

Platform tokens were updated. "Instagram" dropped while its abbreviation "ig" flourished. In both samples "snapchat" declined. Users became more likely to invite or discourage direct messages ("dm").

Preferred pronouns were again the biggest winners. Perhaps related, the rainbow flag emoji (a symbol of LGBTQ pride or solidarity) was another winner.

I remind the reader: these are short lists of the tokens most dramatically changing in prevalence. There are annual estimates for thousands of other tokens in the database, which anyone may download. For a more immediate look into the data–also available to anyone–I describe below a web-based visualization tool.

**Table 5. Top 20 loser tokens in the longitudinal sample.**

| Token | Annual Prevalence Change Estimate | Standard Error of Estimate |
|---|---|---|
| i | -49 | 2.68 |
| a | -29 | 1.64 |
| love | -21 | 0.56 |
| life | -19 | 1.40 |
| you | -19 | 1.88 |
| me | -19 | 1.13 |
| snapchat | -18 | 5.77 |
| with | -13 | 0.28 |
| Red heart emoji* | -12 | 3.07 |
| student | -11 | 0.54 |
| 15 | -11 | 1.46 |
| music | -11 | 0.23 |
| am | -10 | 0.91 |
| im | -10 | 2.53 |
| 17 | -9 | 2.70 |
| 18 | -9 | 2.45 |
| like | -9 | 0.49 |
| 16 | -9 | 2.28 |
| follow | -9 | 1.52 |
| that | -8 | 0.92 |

This is a curated list of the 20 tokens with the largest estimated annual decrease in prevalence. Punctuation, individual digits, a small set of stop words (e.g. of) and URL components (e.g. https) have been previously filtered. Emoji results are marked with *.

## The online tool: Jason J. Jones Identity Trends V1

Jason J. Jones Identity Trends V1 is a website that anyone can use to explore the dataset without programming or any software beyond a web browser. It is available at https://jasonjones. ninja/jason-j-jones-identity-trends-v1/. To explore, one types 1–10 tokens of interest and chooses either the cross-sectional or longitudinal sample. Fig 5 is a screenshot of the interface.

The tool will return the results as both a graph and a table. Fig 6 is a screenshot of results for the tokens "mother, father, mom and dad" in the longitudinal sample. The graph shows that every token except "mother" increased in prevalence. One can also infer that male parents prefer to present themselves as "father" more than "dad," while female parents prefer "mom" over "mother."

Of course, the tool cannot tell you why prevalence differs between tokens or why it changes over time. That is an exercise left to the user of the tool. What it does provide is precise estimates of prevalence based on observations over millions of individuals across six years.

The tool is partially inspired by Google Trends, which has been mentioned by name in thousands of research articles indexed by Google Scholar, and was the main source of data for hundreds of articles reviewed by [16]. Google Trends provides a useful measure of relative interest in search terms over time and geography. Similarly, Jason J. Jones Identity Trends V1 aims to deliver useful estimates of tokens' prominence within individuals' self-conception, or at least self-presentation.

Like any tool, JJJIT V1 has limitations. Currently, it is limited to US users of Twitter. This is due to convenience; Twitter makes public data easy to collect, and the author is interested in

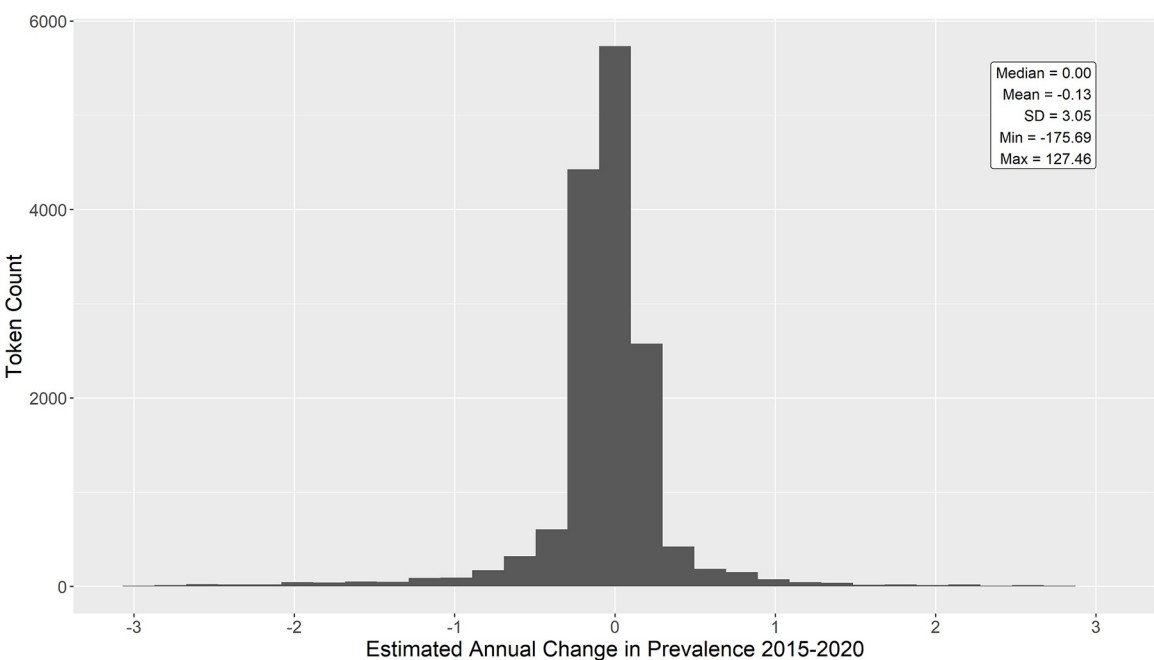

**Fig 4. Distribution of estimated annual change in prevalence for all unique tokens within the cross-sectional sample.** Prevalence is stable (i.e. zero change) for many tokens. Note from the min and max annotations that extreme values are present in the data but not pictured here. See Tables 6 and 7 for illustration.

**Table 6. Top 20 winner tokens in the cross-sectional sample.**

| Token | Annual Prevalence Change Estimate | Standard Error of Estimate |
|---|---|---|
| she | 30 | 10.19 |
| her | 27 | 9.24 |
| he | 16 | 5.81 |
| him | 16 | 5.04 |
| maga | 16 | 1.66 |
| 22 | 15 | 3.57 |
| 21 | 14 | 3.82 |
| American flag emoji* | 14 | 1.00 |
| trump | 14 | 0.78 |
| twitch | 13 | 1.19 |
| mom | 13 | 1.18 |
| 2020 | 12 | 1.80 |
| here | 11 | 1.60 |
| ig | 10 | 3.84 |
| 23 | 10 | 2.81 |
| dm | 10 | 1.17 |
| account | 10 | 1.35 |
| father | 8 | 1.15 |
| blm | 8 | 3.43 |
| Rainbow flag emoji* | 8 | 0.94 |

This is a curated list of the 20 tokens with the largest estimated annual increase in prevalence in the cross-sectional sample. Punctuation, individual digits, a small set of stop words (e.g. of) and URL components (e.g. https) have been previously filtered. Emoji results are marked with *.

**Table 7. Top 20 loser tokens in the cross-sectional sample.**

| Token | Annual Prevalence Change Estimate | Standard Error of Estimate |
|---|---|---|
| a | -53 | 5.59 |
| i | -39 | 6.95 |
| you | -33 | 6.41 |
| life | -31 | 2.78 |
| love | -25 | 2.79 |
| with | -25 | 1.41 |
| music | -22 | 0.50 |
| snapchat | -19 | 4.79 |
| we | -17 | 0.85 |
| your | -16 | 1.70 |
| me | -16 | 6.22 |
| that | -14 | 1.84 |
| follow | -13 | 3.97 |
| 15 | -13 | 2.30 |
| live | -12 | 1.18 |
| one | -11 | 1.37 |
| im | -11 | 3.12 |
| Red heart emoji* | -11 | 3.11 |
| marketing | -11 | 0.72 |
| instagram | -11 | 1.11 |

*Note*: This is a curated list of the 20 tokens with the largest estimated annual decrease in prevalence in the cross-sectional sample. Punctuation, individual digits, a small set of stop words (e.g. of) and URL components (e.g. https) have been previously filtered. Emoji results are marked with *.

US social trends. With further development, coverage could be expanded to other social networking sites and countries. Currently, the tool can only provide estimates from 2015 through 2020. Data is continuously compiled, and 2021 estimates will become available. Historical data, if made available, could extend estimates back to 2007. Currently, one may only query unigrams (i.e. single tokens). With further development, n-grams of varying length could be supported. Currently, the tool only provides estimates at annual resolution. Version 2 of the tool will do so at daily resolution.

## Conclusions

Studying personally expressed identity at scale provides unprecedented opportunity for analysis. The demonstration analysis above merely dips a hesitant toe into the ocean of available data. Already, there is replication of previous work [10, 17]. A prominent trend–pronoun-slash-lists–emerged and calls loudly for study as a social contagion. Other instances of norm diffusion and evolving self-representation certainly dwell outside of the few tokens considered here.

No instrument has or likely ever will reach the scale and resolution of the LOPS method. Personally expressed identity can and should be studied with data from millions of individuals at fine temporal and geographic resolution. The dataset, theory, and methods offered in this article present an approach that has already supported new works. The possibility for further development stretches to all horizons of time, space and the self.

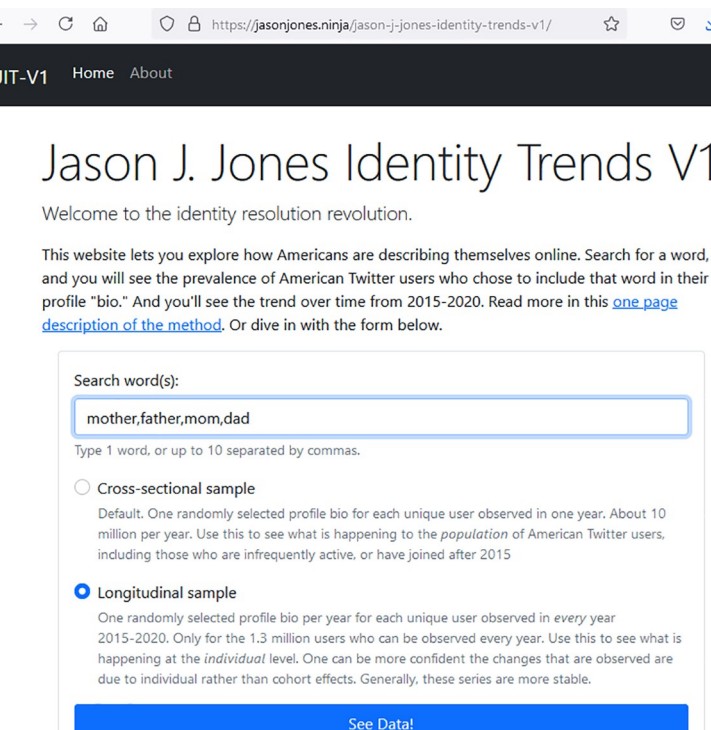

**Fig 5. The interface for the web tool: Jason J. Jones Identity Trends V1.** The tool is publicly accessible at https://jasonjones.ninja/jason-j-jones-identity-trends-v1/.

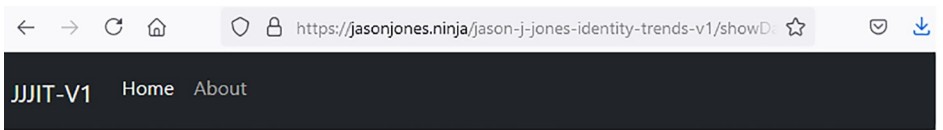

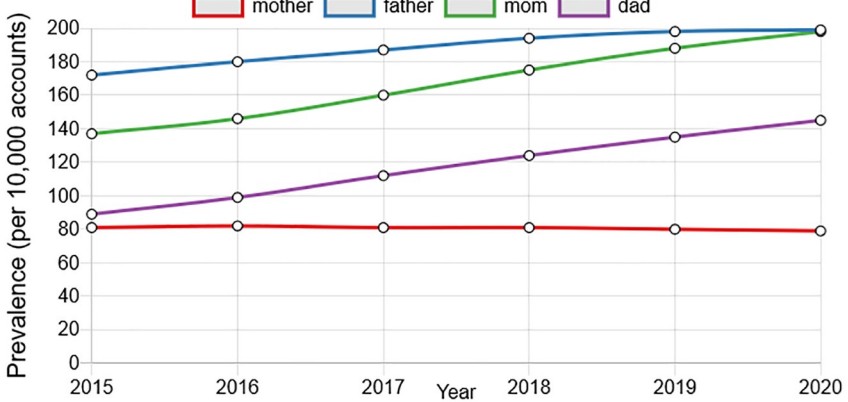

**Fig 6. Example results from the web tool.**

## Supporting information

**S1 File.**
(PDF)

## Author Contributions

**Conceptualization:** Jason Jeffrey Jones.

**Data curation:** Jason Jeffrey Jones.

**Formal analysis:** Jason Jeffrey Jones.

**Funding acquisition:** Jason Jeffrey Jones.

**Investigation:** Jason Jeffrey Jones.

**Methodology:** Jason Jeffrey Jones.

**Project administration:** Jason Jeffrey Jones.

**Resources:** Jason Jeffrey Jones.

**Software:** Jason Jeffrey Jones.

**Visualization:** Jason Jeffrey Jones.

**Writing – original draft:** Jason Jeffrey Jones.

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
