## [Decision Letter · Decision Letter 0]

7 Oct 2021

PONE-D-21-22972A Dataset for the Study of Identity at Scale: Annual Prevalence of American Twitter Users with specified Token in their Profile Bio 2015-2020PLOS ONE

Dear Dr. Jones,

Thank you for submitting your manuscript to PLOS ONE. After careful consideration, we feel that it has merit but does not fully meet PLOS ONE’s publication criteria as it currently stands. Therefore, we invite you to submit a revised version of the manuscript that addresses the points raised during the review process. The revised version should consider the suggestions made by the reviewers. Please note that any references suggested by the reviewers should be included only if they are truly relevant for the paper.

We look forward to receiving your revised manuscript.

Kind regards,

Liviu-Adrian Cotfas

Academic Editor

PLOS ONE

Journal Requirements:

3. Please remove your figures from within your manuscript file, leaving only the individual TIFF/EPS image files, uploaded separately.  These will be automatically included in the reviewers’ PDF

4. We note that Figure 1 in your submission contain copyrighted images. All PLOS content is published under the Creative Commons Attribution License (CC BY 4.0), which means that the manuscript, images, and Supporting Information files will be freely available online, and any third party is permitted to access, download, copy, distribute, and use these materials in any way, even commercially, with proper attribution. For more information, see our copyright guidelines: http://journals.plos.org/plosone/s/licenses-and-copyright.

Reviewers' comments:

Reviewer's Responses to Questions

**Comments to the Author**

1. Is the manuscript technically sound, and do the data support the conclusions?

Reviewer #1: Yes

Reviewer #2: Yes

2. Has the statistical analysis been performed appropriately and rigorously? 

Reviewer #1: Yes

Reviewer #2: Yes

3. Have the authors made all data underlying the findings in their manuscript fully available?

Reviewer #1: Yes

Reviewer #2: Yes

4. Is the manuscript presented in an intelligible fashion and written in standard English?

Reviewer #1: Yes

Reviewer #2: Yes

5. Review Comments to the Author

Reviewer #1: The paper presents a valuable contribution regarding the use of online data to study the evolution of social trends in the US over the 2015-2020 time frame. The author precisely describes the methodology used to compile the dataset, the metrics developed to analyze it, and shows a few research avenues that could be investigated through this data.

While the language used by the author is clear and precise, the structure of the article could sometimes be clarified. Some methodological elements are defined several times over the article (such as the calculation of prevalence, page 7 and again page 14). The section on "Longitudinal Online Profile Sampling" appears after the presentation of the dataset. The paper could be improved with a more rigorous separation of sections such as data collection, metrics, online tool, examples ...

The simple tokenisation approach described in the article (separation over word boundaries) seems to provide interesting results. However, some classic linguistic methods such as lemmatization could be investigated to study not only one given form of a term ("kitty", "kitties", "kitten") but all the form of a given term ("kitt-" or even "cat-"). Such process could be an interesting addition to the dataset.

Finally, I have to point out that the name and URL of the online tool infringe the anonymity of the author. When submitting this kind of paper through a peer review process, the name of the tool should be anonymized and the URL should only be integrated in the final version.

Reviewer #2: Review of

A Dataset for the Study of Identity at Scale: Annual Prevalence of American Twitter - Users with specified Token in their Profile Bio 2015-2020

I have read and reviewed ‘A Dataset for the Study of Identity at Scale: Annual Prevalence of American Twitter. Users with specified Token in their Profile Bio 2015-2020’. This article provides information about a new method and data set to study self-reported self-concept via the Twitter bio, and trends therein. The article is written very well and describes an interesting and relevant method for analyzing trends in the way people chose to present themselves.

And in this last sentence lies my only major criticism of the present article: it strikes me as purely descriptive, and would benefit from a little more background with regard to the theme of self-concept. The references the author gives on this topic seem incomplete. I would personally recommend a bit more information about what type of self-representation a Twitter bio would constitute on the basis of literature on the topic of self-categorization and generally the psychology of identity. I would suggest the literature listed (incompletely!) below. Without more information about the psychology of how and why people chose to represent themselves in the way they do, the analysis is rather superficial and only describes trends (‘winners and losers’), in my opinion. To add any sort of meaning to those trends, I think more background is needed.

This would also provide more insight into questions about to what degree the medium, Twitter, influences the way in which people choose to represent themselves.

To be perfectly clear: my remarks are only valid if the author intended to not only introduce the dataset, and the data collection method, but also decribe how this data can be analyzed. If not, than the article is concise and short description of said aspects.

In sum: the author describes an interesting and potentially promising method for collecting self-reported data on identity. The article is well written and the used methods are sound. As a reviewer I feel however, that a more complete discussion of the relevant literature on self-identity, self-categorization and self-representation would help to better appreciate the value of the collected data.

Suggested literature:

Lea, M., Spears, R., & De Groot, D. (2001). Knowing me, knowing you: Anonymity effects on social identity processes within groups. Personality and Social Psychology Bulletin, 27(5), 526-537.

Reicher, S. D., Haslam, S. A., Spears, R., & Reynolds, K. J. (2012). A social mind: The context of John Turner’s work and its influence. European review of social psychology, 23(1), 344-385.

Thumim, N. (2012). Self-representation and digital culture. Springer.

Van Zomeren, M., Postmes, T., & Spears, R. (2008). Toward an integrative social identity model of collective action: a quantitative research synthesis of three socio-psychological perspectives. Psychological bulletin, 134(4), 504.

6. PLOS authors have the option to publish the peer review history of their article (what does this mean?). If published, this will include your full peer review and any attached files.

Reviewer #1: No

Reviewer #2: No

---

## [Author Response · Author response to Decision Letter 0]

2 Nov 2021

October 29, 2021

Dear Editors and Reviewers,

I thank you for your comments on my manuscript (PONE-D-21-22972) titled A Dataset for the Study of Identity at Scale: Annual Prevalence of American Twitter Users with specified Token in their Profile Bio 2015-2020.

In response to some points, I have revised the paper. On other points I respond here in this letter. Here are points made in review and my responses:

Journal Requirements:

I have made edits to ensure the manuscript meets PLOS ONE's style requirements. These include reordering the manuscript and revising headings to conform to PLOS ONE style.

The following statement best describes the grant funding of this research. This statement should be included wherever the journal deems it to be necessary.

This material is based upon work supported by the National Science Foundation under grant IIS-1927227. The author thanks Stony Brook Research Computing and Cyberinfrastructure, and the Institute for Advanced Computational Science at Stony Brook University for access to the high-performance SeaWulf computing system, which was made possible by a $1.4M National Science Foundation grant (#1531492).

3. Please remove your figures from within your manuscript file, leaving only the individual TIFF/EPS image files, uploaded separately. These will be automatically included in the reviewers’ PDF

I have removed the figures from the manuscript file and uploaded the image files separately as requested.

4. We note that Figure 1 in your submission contain copyrighted images. All PLOS content is published under the Creative Commons Attribution License (CC BY 4.0), which means that the manuscript, images, and Supporting Information files will be freely available online, and any third party is permitted to access, download, copy, distribute, and use these materials in any way, even commercially, with proper attribution. For more information, see our copyright guidelines: http://journals.plos.org/plosone/s/licenses-and-copyright.

I have removed Figure 1 from the manuscript. It was included for illustration and not necessary for understanding the manuscript.

I have reviewed the reference list to ensure it is complete and correct. To my knowledge, the article does not and did not cite any retracted work. Please inform me if that is not accurate.

Reviewer #1

The paper presents a valuable contribution regarding the use of online data to study the evolution of social trends in the US over the 2015-2020 time frame. The author precisely describes the methodology used to compile the dataset, the metrics developed to analyze it, and shows a few research avenues that could be investigated through this data.

While the language used by the author is clear and precise, the structure of the article could sometimes be clarified. Some methodological elements are defined several times over the article (such as the calculation of prevalence, page 7 and again page 14). The section on "Longitudinal Online Profile Sampling" appears after the presentation of the dataset. The paper could be improved with a more rigorous separation of sections such as data collection, metrics, online tool, examples ...

I have re-ordered and revised headings in response to this feedback and the request to conform to journal formatting. Specifically, "Longitudinal Online Profile Sampling" section appears in the Materials and Methods section before the presentation of the dataset (in the Results section) as the reviewer requested. Also, the calculation of prevalence is defined only once in this revision.

The simple tokenisation approach described in the article (separation over word boundaries) seems to provide interesting results. However, some classic linguistic methods such as lemmatization could be investigated to study not only one given form of a term ("kitty", "kitties", "kitten") but all the form of a given term ("kitt-" or even "cat-"). Such process could be an interesting addition to the dataset.

I thank the reviewer for the suggestion, but argue this form of analysis does not belong in this manuscript. Modern approaches to Natural Language Processing often skip lemmatization. Lemmatization is useful when one has limited text data, and it is assumed different linguistic forms necessarily relate to the same semantic content. But in my application, language is plentiful, and I am specifically interested in the exact words users choose to represent themselves. 

Finally, I have to point out that the name and URL of the online tool infringe the anonymity of the author. When submitting this kind of paper through a peer review process, the name of the tool should be anonymized and the URL should only be integrated in the final version.

The name of the tool and the URL includes my full name intentionally. Scientists should demand and receive more credit for their work. This journal does not require blind review. Even in venues that promise blind review, that promise is rarely delivered upon. A Google search on the title or any unique phrasing would reveal the identity of the author of this work (me), and that is something I cannot control.

Reviewer #2: 

I have read and reviewed ‘A Dataset for the Study of Identity at Scale: Annual Prevalence of American Twitter. Users with specified Token in their Profile Bio 2015-2020’. This article provides information about a new method and data set to study self-reported self-concept via the Twitter bio, and trends therein. The article is written very well and describes an interesting and relevant method for analyzing trends in the way people chose to present themselves.

And in this last sentence lies my only major criticism of the present article: it strikes me as purely descriptive, and would benefit from a little more background with regard to the theme of self-concept. The references the author gives on this topic seem incomplete. I would personally recommend a bit more information about what type of self-representation a Twitter bio would constitute on the basis of literature on the topic of self-categorization and generally the psychology of identity. I would suggest the literature listed (incompletely!) below. Without more information about the psychology of how and why people chose to represent themselves in the way they do, the analysis is rather superficial and only describes trends (‘winners and losers’), in my opinion. To add any sort of meaning to those trends, I think more background is needed.

The reviewer and I have a difference of opinion. These suggested critiques are in my opinion strengths of the manuscript. “Purely descriptive” is used a pejorative by the reviewer, but in my opinion descriptive work is sorely needed, unduly maligned, and deeply underrepresented in the study of human behavior and especially identity.

“What is the self?” is a fascinating question, but one that has been written about elsewhere at length. I don’t intend to address this question directly in this manuscript. This manuscript’s primary purpose is to introduce a dataset. A survey of ideas of the self-concept could only be sprawling or insufficient in that context. More crucially, it would bias readers’ percepetion of the dataset and the method as associated with a particular theory or set of theories. This would be disingenuous and misleading, because the current effort is data-driven and purposely descriptive in nature.

This would also provide more insight into questions about to what degree the medium, Twitter, influences the way in which people choose to represent themselves.

It is not my intention to address how the medium (Twitter) influences self-representation. Twitter is simply the only source of personally expressed identity data for millions of people across time and space. I leave it to other researchers to seek and explain cross-media effects.

To be perfectly clear: my remarks are only valid if the author intended to not only introduce the dataset, and the data collection method, but also describe how this data can be analyzed. If not, than the article is concise and short description of said aspects.

In sum: the author describes an interesting and potentially promising method for collecting self-reported data on identity. The article is well written and the used methods are sound. As a reviewer I feel however, that a more complete discussion of the relevant literature on self-identity, self-categorization and self-representation would help to better appreciate the value of the collected data.

Here the reviewer and I agree. There is much more to be said about self-identity, self-categorization and self-representation than is in this manuscript. I read the reviewer’s included citations with interest. My aim being a concise description of methods and data, however, I choose to not delve into the history of the many conceptualizations of the self. I hope the editor and reviewers agree this is a fine choice for this manuscript.

Thank you for your time and effort in considering this manuscript.

Please address all correspondence concerning this manuscript to me: jason.j.jones@stonybrook,edu. 

Sincerely,

Dr. Jason J. Jones

Associate Professor

Dept. of Sociology and Institute for Advanced Computational Science

Stony Brook University

---

## [Editor Report · Decision Letter 1]

4 Nov 2021

A dataset for the study of identity at scale: Annual Prevalence of American Twitter Users with Specified Token in their Profile Bio 2015-2020

PONE-D-21-22972R1

Dear Dr. Jones,

We’re pleased to inform you that your manuscript has been judged scientifically suitable for publication and will be formally accepted for publication once it meets all outstanding technical requirements.

Kind regards,

Liviu-Adrian Cotfas

Academic Editor

PLOS ONE
---

## [Editor Report · Acceptance letter]

9 Nov 2021

PONE-D-21-22972R1 

A dataset for the study of identity at scale: Annual Prevalence of American Twitter Users with Specified Token in their Profile Bio 2015-2020 

Dear Dr. Jones:

I'm pleased to inform you that your manuscript has been deemed suitable for publication in PLOS ONE. Congratulations! Your manuscript is now with our production department. 

Kind regards, 

on behalf of

Dr. Liviu-Adrian Cotfas 

Academic Editor

PLOS ONE